# Machine Learning Predicts Outcomes of Phase III Clinical Trials for Prostate Cancer

**Felix D. Beacher** [1,*], **Lilianne R. Mujica-Parodi** [2], **Shreyash Gupta** [1] 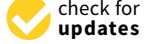 and **Leonardo A. Ancora** [1,3]

[1] Cool Clinical Consortium for AI and Clinical Science, 1092 Lausanne, Switzerland; shreyash@coolclinical.com
[2] Department of Biomedical Engineering, Renaissance School of Medicine, Stony Brook University, Stony Brook, NY 11790, USA; lilianne.strey@stonybrook.edu
[3] Faculty of Medicine, University of Lisbon, 1649-028 Lisbon, Portugal; lancora@edu.ulisboa.pt
[*] Correspondence: felix@coolclinical.com; Tel.: +4178-667-9695

**Abstract:** The ability to predict the individual outcomes of clinical trials could support the development of tools for precision medicine and improve the efficiency of clinical-stage drug development. However, there are no published attempts to predict individual outcomes of clinical trials for cancer. We used machine learning (ML) to predict individual responses to a two-year course of bicalutamide, a standard treatment for prostate cancer, based on data from three Phase III clinical trials ($n = 3653$). We developed models that used a merged dataset from all three studies. The best performing models using merged data from all three studies had an accuracy of 76%. The performance of these models was confirmed by further modeling using a merged dataset from two of the three studies, and a separate study for testing. Together, our results indicate the feasibility of ML-based tools for predicting cancer treatment outcomes, with implications for precision oncology and improving the efficiency of clinical-stage drug development.

**Keywords:** clinical trials; machine learning; classification; prostate cancer; precision medicine; drug development

## 1. Introduction

'Precision medicine' refers to the tailoring of medical treatment depending on a patient's individual characteristics, e.g., genes, environmental factors, and lifestyle. If physicians could accurately predict an individual patient's responses to different treatment options, the best option could be selected. For most diseases, the efficacy and safety of standard treatments are highly variable. Thus, individual-specific treatment protocols have been hailed as an emerging revolution in medicine, with the potential to improve patient care and deliver cost savings to health services [1].

Currently, precision medicine in real-world clinical practice is mainly associated with treatment based on cancer subtype and genotype. For example, olaparib is a monotherapy for ovarian cancer in women with BRCA1/2 mutations [2]. However, there are still few examples of real-world precision medicine. Current clinical practice still relies heavily on subjective judgment and limited individual patient data [3]. A 'one-drug-fits-all' approach is often used, in which a particular diagnosis leads to a specific type of treatment. Alternatively, trial-and-error practices are common, in which various treatment options are tried in the hope that one will work.

Machine learning (ML) has been described as 'the key technology' for the development of precision medicine [4]. ML uses computer algorithms to build predictive models based on complex patterns in data. ML can integrate the large amounts of data required to "learn" the complex patterns required for accurate medical predictions. ML has excelled in diagnostics, e.g., in neurodegenerative diseases [5], cardiovascular disease [6], and cancer [7]. ML approaches have also been used to predict treatment outcomes for a range of conditions, including schizophrenia [8], depression [9], and cancer [10].

In 2014, IBM launched 'Watson for Oncology,' which aimed to use ML to recommend treatment plans for cancer, based on combined inputs from research, a patient's clinical notes, and the clinician [11]. However, this project has so far failed to deliver the kinds of commercial products which had been envisioned [12]. Other reports have used ML to predict treatment outcomes for cancer. One study used ML to predict patient survival based on microscopy images of cancer biopsy tissue and genomic markers [13]. Another study used ML to predict response to treatment in patients with cervical and head cancers based on PET images [14]. Despite such advances, there are currently no ML-based tools approved by regulators.

A significant limitation of these kinds of exploratory approaches to ML-based tools for clinical practice is that they tend to rely on types of data that are expensive to collect and may require specialist skills to analyze (e.g., genomics or MRI/PET imaging). This limitation can be a barrier to translating systems into tools for routine clinical practice. One possible way to address this is to base such systems on data from Phase III clinical trials: studies that provide the pivotal data for regulators to assess whether a new drug should be commercially approved. Phase III clinical trials are typically large enough for ML (usually 1000+ subjects) and include a wide range of data (e.g., demographic, clinical, and biochemical), which can be easily stored in tabular format. Moreover, the current trend is towards making clinical trial datasets publicly available. Thus, Phase III clinical trial data may be a good source for developing practical ML-based tools for precision medicine. This approach is relatively untried: a literature search revealed only one prior study that used clinical trial data to predict treatment responses [15]. This study used ML to predict responses to an anti-depressant after 12 weeks. This was an important first step, however the accuracy level was modest (65%). It is possible that, with different data, and with a different approach to modeling, a level of accuracy could be achieved which could lead to the development of tools for real-world clinical practice.

In addition to aiding precision medicine, the ability to predict individual treatment responses in clinical trials could improve the efficiency of clinical-stage drug development. This is an important issue because the average development cost of a new drug has become extremely expensive in recent decades, estimated at $2.6 billion (2013 U.S. dollars; [16]). In addition, investigational drugs in clinical trials have a significant rate of rejection by regulators [17]. Together, these factors represent a significant threat to existing models of drug development. It has even been claimed that we are entering a 'post-blockbuster era' [18], in which the development costs of new drugs are becoming prohibitive.

The ability to predict clinical trial outcomes could cut the costs of clinical trials (by allowing smaller sample sizes) and improve the chance of regulatory approval by improving efficacy and safety within a pre-selected patient subpopulation predicted to benefit from the drug. This strategic pre-selection is known as 'clinical trial enrichment.' There have not yet been any regulatory approvals of drugs using clinical trial enrichment based on ML. However, the U.S. Food and Drug Administration (FDA) has encouraged research in this area [19]. Accordingly, ML-based clinical trial enrichment could become an essential part of drug development.

Prostate cancer is a leading cause of cancer deaths in men [20]. The two main first-line treatments for prostate cancer are prostatectomy and radiotherapy to the prostate. The typical second-line treatment for prostate cancer is androgen deprivation therapy (ADT), which results in dramatic reductions in androgen levels [21]. ADT can mean either surgical castration or treatment with pharmacological antiandrogens.

ADT typically causes demasculinization side effects, such as reduced sexual function, genital shrinkage, and loss of muscle mass. Further, ADT may increase the risk for high blood pressure, diabetes, stroke, and heart disease [22]. Finally, cancer relapse after 2–3 years of ADT is prevalent. These risks and uncertainties underline the need to develop precision medicine approaches for prostate cancer. One common pharmacological form of ADT is bicalutamide, which is approved as a monotherapy in over 50 countries. This current study used Phase III clinical trial data on bicalutamide as a treatment for prostate

cancer (drug arm only; see Materials and Methods for details). A standard evaluation of efficacy and tolerability using data from the same studies (drug arm vs. placebo) was published in 2010 [23]. This study reported that bicalutamide treatment significantly improved progression-free survival in patients with advanced prostate cancer, but not in patients with only localized disease.

Early-stage prostate cancers are often detected by routine measurement of blood levels of prostate-specific antigen (PSA). PSA levels are strongly correlated with tumor diagnosis, tumor aggressiveness, and bone metastasis [24] and are a recognized disease biomarker. Change in PSA levels was critical to how we operationalized the response to bicalutamide in this study.

To date, there have been no published attempts to predict individual cancer treatment outcomes based on data from clinical trials. This is an important gap in the literature. To address this gap, we processed data from three comparable Phase III clinical trials on bicalutamide for prostate cancer and used ML to predict two-year treatment outcomes, based only on measurements available at baseline.

## 2. Materials and Methods

### 2.1. Sources of Data

The sources of data for this study were the Astra-Zeneca Early Prostate Cancer clinical program. This program consisted of three Phase III clinical trials on bicalutamide or placebo in subjects with prostate cancer. These three studies had slightly different procedures but were designed to yield data that could be pooled. The data and study protocols were obtained through Project Data Sphere (PDS UID: Prostat_AstraZe_1995_102, Prostat_AstraZe_1995_105, Prostat_AstraZe_1995_106). The datasets publicly released contain data for the drug arm but not the placebo arm. Basic information about these studies is given in Table 1.

**Table 1.** Summary of the three Bicalutamide prostate cancer studies in the Astra-Zeneca Early Prostate Cancer program, the sources of data for this study.

|  | Study 1 | Study 2 | Study 3 |
|---|---|---|---|
| **Title** | A Randomized Double-Blind Comparative Trial of Bicalutamide (Bicalutamide) Versus Placebo in Patients with Early Prostate Cancer | A Randomised, Double-blind, Parallel-group Trial Comparing Bicalutamide 150 mg Once Daily With Placebo In Subjects With Non-metastatic Prostate Cancer | A Randomised, Double-blind, Parallel-group Trial Comparing Bicalutamide 150 mg Once Daily With Placebo In Subjects With Non-metastatic Prostate Cancer |
| **Sites** | U.S. and Canada | Mostly Europe, but also Mexico, Australia and South Africa | Sweden, Norway, Finland, Denmark |
| **Year first subject enrolled** | 1995 | 1995 | 1995 |
| **Year study completed** | 2008 | 2008 | 2008 |

### 2.2. Objectives of the Original Studies

The stated objectives of these studies varied slightly, but were similar and can be summarized as:

1. To compare time to clinical progression after two years of bicalutamide 150 mg monotherapy vs placebo.

2. To compare overall survival after two years of bicalutamide 150 mg monotherapy vs placebo.
3. To evaluate tolerability of two years of bicalutamide 150 mg therapy versus placebo.
4. To compare treatment failure after two years of bicalutamide 150 mg monotherapy vs placebo.

### 2.3. Design of the Original Studies

The three studies all had a randomized, double-blind, parallel-group design, comparing bicalutamide with placebo. Subjects were randomized 1:1 to bicalutamide or placebo. The study design is illustrated in Appendix A.

### 2.4. Subjects and Datasets

Inclusion and exclusion criteria were similar for the three studies, however, there were some differences (see Appendix B). Demographic and clinical features of subjects are shown in Table 2.

**Table 2.** Demographics, and clinical features of subjects, based on the subjects available for final analysis.

| | | Study 1 | Study 2 | Study 3 |
|---|---|---|---|---|
| | | (*n* = 1625) | (*n* = 1440) | (*n* = 588) |
| **Demographics** | Mean age (years) | 64 | 69 | 68 |
| | Race | Cauc 84% (1372) | Cauc 95% (1365) | Cauc 99% (584) |
| | | Black 11% (188) | Black 1% (13) | Black 0% (0) |
| | | Asian 0.7% (12) | Asian 0.5% (7) | Asian 0% (0) |
| | | Hispanic 3% (46) | Hispanic 2% (25) | Hispanic 0.3% (2) |
| | | Other 0.4% (7) | Other 2% (31) | Other 0.3% (2) |
| **Clinical features** | Clinical stage | | | |
| | Category 1 | 0.2% (3) | 0.1% (1) | 0% (0) |
| | Category 2 | 1% (23) | 11% (153) | 7% (44) |
| | Category 3 | 8% (131) | 17% (245) | 15% (89) |
| | Category 4 | 63% (1028) | 38% (550) | 39% (229) |
| | Category 5 | 27% (440) | 31% (446) | 36% (213) |
| | Category 6 | 0% (0) | 3% (45) | 2% (13) |
| | Metastatic node disease | 28% (462) | 3% (41) | 4% (24) |
| | Gleason score category | | | |
| | Category 1 | 5% (78) | 33% (477) | 44% (256) |
| | Category 2 | 48% (788) | 40% (574) | 45% (266) |
| | Category 3 | 47% (759) | 25% (363) | 11% (63) |
| | Category 4 | 0% (0) | 2% (26) | 0.5% (3) |

The studies assessed a wide range of information, including demographics, clinical information (such as clinical stage, and the presence of metastatic disease), physical characteristics (such as weight), and concomitant medications.

### 2.5. Comparability of the Studies

The comparability of the studies is important to assess the viability of pooling the data across studies. The three studies were similar in terms of:

- Age.
- Racial makeup (predominantly white, although differences in frequencies of minority races).
- Clinical stage.
- Gleason score category.
- Proportion of serious related A.E.s.
- Proportion of death within two years.

However, there were differences between the three studies in some variables, for example:

- Frequency of prior prostatectomy, reflecting different inclusion criteria (see Appendix B).
- Frequency of metastatic node disease at baseline.
- Mean PSA levels over the two-year main trial period (likely related to the greater incidence of prior prostatectomy or prostate radiation treatment before randomization).

There were some differences in data collection methods and assessments. These differences are relevant because it is only assessments in common that could be used in modeling across studies. In some cases, differences in assessments could be resolved by converting variables from one form to another (e.g., Gleason scores could be converted into Gleason categories).

### 2.6. Derivation of the Target Variable

The classification problem dealt with two classes. A patient could either be 'good responder' or a 'bad responder'. This target variable was defined such that it reflected both efficacy and safety/drug toleration. For all the datasets, there were insufficient cases of death and clinical progression for these to form the basis of the target variable. Therefore, efficacy was defined in terms of PSA levels, a recognized biomarker for prostate cancer disease progression.

A subject was a 'good responder' if he met the following conditions:

1.  PSA levels did not increase by more than 0.2 ng/mL within the initial two-year trial period (efficacy), similar to the method used in the literature [25].

    ○  'PSA increase' was defined in terms of a comparison of baseline vs. 2 years. For Studies 2 and 3, 'baseline' was the day of randomization. For Study 1, 'baseline' was set at three months, to address the observation that in Study 1 (but not Studies 2 or 3) mean PSA values underwent a marked reduction (43%) from day 0 to 3 months and then, on average, stabilized. Thus, this baseline shift was an adjustment designed to ensure consistency between the three studies (i.e., the start of stable PSA values). This discrepancy between studies in the trajectory of PSA values is likely related to different entry criteria. In Study 1, subjects were required to have undergone radical prostatectomy or prostate radiation treatment (which would have caused a marked decrease in PSA levels) within the 16 weeks before randomization, a requirement not present for studies 2 or 3.

    ○  PSA values at 2 years after randomization was operationalized by taking the mean of PSA values within a window of 600–900 days after randomization. The width of this window was designed to balance the need to include a sufficient number of data points, with the need to confine the window, so that the time points could be considered consistent.

2.  There were no related, serious adverse events (A.E.s) within the initial two-year trial period (safety)

    ○  A related A.E. was considered serious if it:

    1.  caused withdrawal from the study OR

2. caused disability or incapacity OR
3. required or prolonged hospitalization OR
4. required intervention to prevent impairment OR
5. was life-threatening

3. Survived for the initial two-year trial period (safety)
4. Stayed in the trial for at least 600 days (safety)

If a patient did not meet all four of these conditions, he was considered to be a "bad responder".

Subjects were excluded from the analyses if there were insufficient data to evaluate all of these conditions.

### 2.7. Feature Reduction

'Feature reduction' (i.e., selection of predictor variables) can improve the accuracy of ML models by removing noise from the training data. In the datasets used in this study, a particular issue was that binary coding of concomitant medications yielded over 5000 features. Therefore, feature reduction was applied. For all datasets, binary variables with less than 10 positive cases (i.e., very low variance) were excluded.

For models using datasets merged from more than one study, only features in common for all three studies were included. Further feature reduction was applied according to the following steps:

1. The Chi-squared test was used to identify which categorical variables were significantly related to the target variable ('good responders'; $p < 0.05$). Unrelated variables were removed.
2. ANOVA was used to test which continuous variables were significantly different between the target classes ($p < 0.05$). Unrelated variables would have been removed, but none needed to be removed on this basis.
3. Where there were high correlations between the remaining continuous predictors (Pearson's $r > 0.9$), the predictor with the lower correlation with the target was removed.

### 2.8. Other Data Preprocessing Steps

Other main preprocessing steps were:

1. Extraction of data from original sas files and compilation into integrated datasets with one row per subject.
2. Recoding of some variables (e.g., Gleason score and clinical stage) to make them consistent between studies.
3. Imputation of missing values (mean for continuous variables, mode for categorical variables).
4. Outlier inspection (there were no implausible values, so none were removed).

Normalization is a standard requirement for machine learning algorithms. Normalization of predictor data was conducted using a scaler robust to outliers. This scaler is known as 'RobustScaler' as implemented in the Python library Scikit-learn. The procedure, for each feature, centers data on the median and scales the data by dividing them by the interquartile range. The resulting features have a mean and median of zero and a standard deviation of 1.

### 2.9. Final Datasets

Preprocessing resulted in the creation of five datasets. The features and descriptive statistics of the datasets merged from the three studies are given in Appendix C.

A reasonable level of balance between the frequency of the classes in the target variable is important to optimize the accuracy of ML predictions. The individual study datasets were highly imbalanced in terms of the frequencies of the target variable classes, but the datasets merging studies were well balanced in this respect: the dataset merging data from studies 1, 2, and 3 had 46% good responders; the dataset merging data from studies 1 and

3 had 56% good responders (Table 3). Both of these proportions are close to the ideal 50% split between the target variable classes.

**Table 3.** Sample sizes and class imbalance for the five datasets used for modeling.

|  | Study 1 | Study 2 | Study 3 | Merged Studies 1,2,3 | Merged Studies 1, 3 |
|---|---|---|---|---|---|
| **N** | 1625 | 1440 | 588 | 3653 | 2213 |
| **% good responders** | 71% | 30% | 16% | 46% | 56% |
| **Number of features** | 93 | 86 | 93 | 23 | 23 |

The separate study datasets included additional features, which were not in common for all three studies. Study 1 also included the indications for which concomitant medications had been prescribed. Studies 2 and 3 also included medical history and physical abnormalities based on a physical examination. Study 3 also included the Golombok Rust Inventory of Sexual Satisfaction questionnaire [26], a 28-item questionnaire for the assessment of sexual dysfunction in heterosexual couples. The full datasets are available on request.

*2.10. Planned Modeling*

All models involved the prediction of 'good responders,' based only on data available at baseline. We developed three types of models:

1.　Three-Study Merged Dataset Models

These models used a dataset that merged data from all three studies. The expectation was that the larger sample size would yield greater accuracy than the other models. These models are applicable to the development of systems for precision medicine. However, they may be less relevant to predictive clinical trial enrichment since the acquisition of datasets of this size will often be unfeasible in the real world.

2.　Separate Study Validation Models

For these models, datasets for Studies 1 and 3 were merged for model training, with the dataset for Study 2 used for model testing. These models provide a test of external generalizability. Studies 1 and 3 were chosen for model training because this merged dataset was well balanced in terms of target variable classes, with 56% good responders. However, the dataset for Study 2 was less well balanced, with 30% good responders.

3.　Individual Study Models

These models used datasets from studies 1, 2, and 3, used independently. This allowed the development of models which could be used to informally explore relationships between model performance, sample size, and class distribution (i.e., the proportions of good and bad responders).

These model types are illustrated in Figure 1.

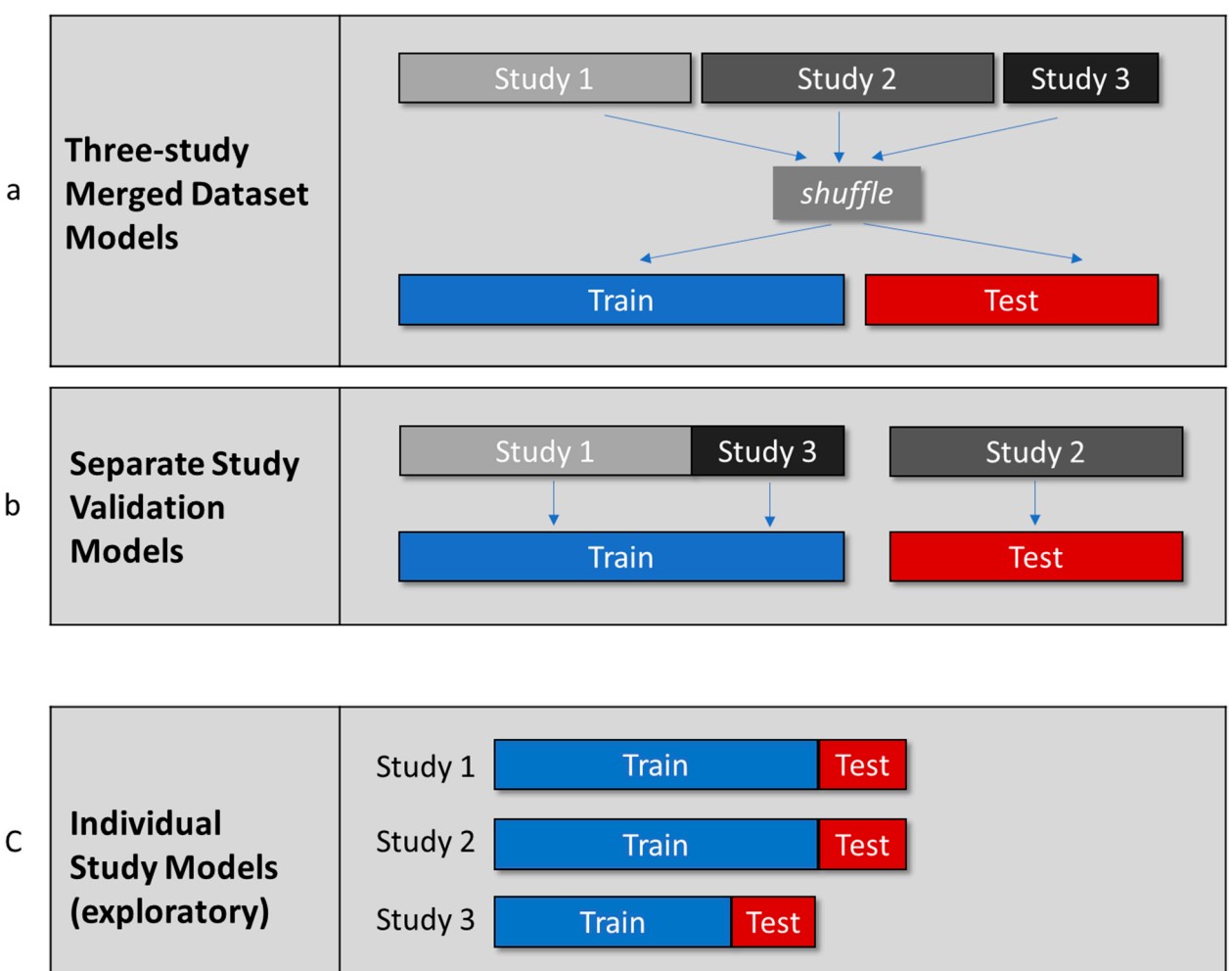

**Figure 1.** The three types of models developed in this study, including: (**a**) Three-study Merged Dataset Models; (**b**) Separate Study Validation Models; (**c**) Individual Study Models. The lengths of the bars approximately indicate the size of the datasets.

*2.11. Classification Algorithms Used*

Five ML algorithms were used, based on their wide recognition in the ML literature, their appropriateness for tabular data and their successful application in previous biomedical studies. For example, XGBoost has been used to predict biomarker outcomes for glioblastomas [27] and K-Nearest Neighbors has been used to predict cognitive decline in Alzheimer's patients [28].

2.11.1. Logistic Regression

Logistic regression is named after the logistic regression transformation function, an S-shaped function that relates some relevant measurement to probabilities of a binary outcome. The goal of logistic regression is to find coefficients for a logistic regression transformation function that minimizes the error between predicted and actual outcomes. Logistic regression is a simple form of ML, which we included as a benchmark.

2.11.2. K-Nearest Neighbors (KNN)

The K-nearest neighbors (KNN) algorithm predicts the class of a new case by assuming it will be the same class as the nearest cases, based on feature distances. The number of cases used in this computation (k) is specified by the user. Like logistic regression, KNN is a simple algorithm, which we included as a second benchmark.

### 2.11.3. CatBoost

The CatBoost ('category boosting') algorithm uses gradient boosting, which makes predictions based on an ensemble of decision trees. While CatBoost tends to perform better with categorical data, it can also handle continuous variables. CatBoost introduced a new system of model regularization called minimal variance sampling, which uses the distribution of loss derivatives and assigns probabilities for sampling of boosting trees, maximizing accuracy.

### 2.11.4. Extreme Gradient Boosting (XGBoost)

XGBoost ('extreme gradient boosting') is a type of gradient boosting algorithm, which trains a series of decision trees to an ensemble, with each new tree designed to correct the errors made by prior models. Weaknesses in the decision trees are identified by minimizing gradients in the loss function, a measure of a model's goodness of fit. XGBoost is an open-source implementation of the gradient boosting algorithm.

### 2.11.5. Voting Classifier

Voting classifiers are an example of ensemble learning in which the predictions from multiple ML algorithms are combined. Final predictions are based on either a majority vote ('hard voting') or the average probability for each prediction ('soft voting'). We developed a voting classifier, based on catboost, light gradient boosting machine (lightGBM), and XGBoost as component classifiers.

### *2.12. Modeling Procedures*

Some model hyperparameters were selected by automated procedures to optimize classification accuracy. 'Grid search' was used if the search space was small and 'random search' was used if the search space was large. All other hyperparameters were set to the default values.

For the logistic regression models, the tuned hyperparameters were: solver, inverse of regularization strength (c) and penalty.

For the KNN models, the tuned hyperparameters were leaf size, number of neighbors and power for the Minkowski metric (p).

For the XGBoost models, the tuned hyperparameters were: minimum child weight, subsample ratio of columns for constructing each tree ('colsample_bytree'), fraction of samples to be used for fitting the individual base learners (subsample), regularization alpha, regularization lambda, number of estimators, maximum depth and learning rate.

For the CatBoost models, the tuned hyperparameters were: depth, learning rate, and number of iterations.

The Voting Classifier used three component classifiers: XGBoost, CatBoost, and LGBoost. For the XGBoost and CatBoost the hyperparameters were the same as those listed above. For the LGBoost, the tuned hyperparameters were: minimum child weight, subsample ratio of columns for constructing each tree ('colsample_bytree'), fraction of samples to be used for fitting the individual base learners (subsample), regularization alpha, regularization lambda, number of estimators, maximum depth, learning rate, number of leaves, and minimum number of data points needed in a leaf ('min_child_samples').

The three-study merged dataset models and the individual study models were assessed with stratified ten-fold cross-validation. Performance metrics were averaged across the test folds, to gain a more robust measurement of model accuracy. All analyses were implemented in Python (version 3.8). All source code used is available upon request.

## 3. Results

### *3.1. Three-Study Merged Dataset Models*

Key results for the Three-Study Merged Dataset Models are given in Table 4.

**Table 4.** Performance Metrics (mean %s) for the Three-Study Merged Dataset Models. Chance accuracy was 54%.

|  | Accuracy | Precision | Sensitivity | F1 | AUC |
|---|---|---|---|---|---|
| **Voting classifier** | 76% | 74% | 79% | 80% | 73% |
| **XGBoost** | 76% | 74% | 78% | 79% | 73% |
| **Catboost** | 75% | 74% | 76% | 77% | 72% |
| **KNN** | 71% | 71% | 72% | 72% | 66% |
| **Logistic Regression** | 70% | 71% | 72% | 71% | 65% |

Differences in accuracy between models were explored. For this, the three-study merged dataset was split into training and test sets using an 80:20 ratio. The models were trained on the training dataset and accuracy on the test set ($n = 731$) was compared for the five classifiers. Exact McNemar tests (two-tailed) were then used to make all ten possible pairwise comparisons. To adjust for the number of comparisons, a p threshold of 0.01 was used. Using this methodology, there were no significant differences in accuracy between the three most accurate models (voting classifier, XGB and CatBoost). Likewise, there were no significant differences in accuracy between the two least accurate models (KNN and logistic regression). However, the three most accurate models (XGB, voting classifier and CatBoost) were all significantly more accurate than the two least accurate models (KNN and logistic regression).

Hyperparameter tuning for the XGBoost resulted in the use of the following settings: number of estimators was 300, maximum depth = 5, learning rate was 0.01, base score was 0.5, the booster was the gradient boosting tree, minimum child weight was 5, regularization lambda was 0.8, regularization alpha was 0.3, column sample by tree was 0.5, and subsample was 0.8.

Gini importance was computed for each feature, on the basis of the XGBoost model. Table 5 gives these importances, and also class distributions for each feature, which can aid the interpretation of these feature importances. The feature with the highest Gini importance was prostatectomy prior to baseline, which was much more common in good responders (70%) than bad responders (37%).

A receiver operating characteristic (ROC) curve plots the tradeoff between the true positives rate and the false positives rate. The true positive rate is the proportion of positive cases correctly classified (also called 'sensitivity' or 'recall'). The false positive rate is the proportion of negative cases incorrectly classified as positive. The ROC curve for the Three-Study Merged Dataset Models is shown in Figure 2.

### 3.2. Separate Study Validation Models

Key results for the Separate Study Validation Models are given in Table 6.

Differences in accuracy between models were explored, for the Separate Study Validation Models, with Study 2 data being used as the test set ($n = 1440$). Exact McNemar tests (two-tailed) were used to make all ten possible pairwise comparisons. To adjust for the number of comparisons, a p threshold of 0.01 was used. There were no significant differences in accuracy between the three most accurate models (voting classifier, XGB, and CatBoost). Likewise, there were no significant differences in accuracy between the two least accurate models (KNN and logistic regression). However, the voting classifier and CatBoost were significantly more accurate than the two least accurate models (KNN and logistic regression). XGBoost was significantly more accurate than KNN, but the accuracies of XGBoost and logistic regression were not significantly different.

**Table 5.** Gini importances for each feature (based on XGBoost) and comparisons of these features (proportions and measures of central tendency) for good and bad responders, for the Three-Study Merged Dataset.

|  | Gini Importance | Good Responders (*n* = 1683) | Bad Responders (*n* = 1970) |
|---|---|---|---|
| Previous prostatectomy | 0.19 | 70% | 37% |
| Previous radiotherapy | 0.17 | 20% | 15% |
| PSA at baseline (median, ng/mL) [1] | 0.14 | 0.2 | 3 |
| PSA prior to baseline (median, ng/mL) [1] | 0.09 | 10 | 17 |
| Metastatic node disease | 0.05 | 18% | 11% |
| Vitamins, not specified | 0.04 | 11% | 4% |
| Clinical stage (proportion of median category) | 0.04 | 4% | 4% |
| Diazepam | 0.03 | 0% | 1% |
| Enalapril | 0.02 | 3% | 5% |
| Anti-anginal vasodilators, organic nitrates | 0.02 | 4% | 8% |
| Glyceryl trinitrate | 0.02 | 2% | 4% |
| Vitamin K antagonists | 0.02 | 2% | 3% |
| Age (mean, years) | 0.02 | 65 | 68 |
| Aspartate aminotransferase levels (mean, U/L) | 0.02 | 19 | 21 |
| Alanine aminotransferase levels (mean, U/L) | 0.02 | 20 | 22 |
| Gleason score category (median) | 0.02 | 2 | 2 |
| Weight (mean, kg) | 0.01 | 83 | 81 |
| HMG COA reductase inhibitors | 0.01 | 9% | 6% |
| Aspirin | 0.01 | 20% | 16% |
| Propionic acid derivatives | 0.01 | 7% | 4% |
| Furosemide | 0.01 | 2% | 3% |
| Inhalational select beta2-adrenoceptor agonists | 0.01 | 3% | 4% |
| Lisinopril | 0.01 | 4% | 3% |

[1] median was used as a measure of central tendency rather than mean due the presence of extreme values.

**Table 6.** Performance Metrics (mean %s) for the Separate Study Validation Models (averages are given for sensitivity, precision and F1 scores, weighted by the frequency of class labels).

|  | Accuracy | Precision | Sensitivity | F1 | AUC |
|---|---|---|---|---|---|
| **Voting classifier** | 70% | 70% | 69% | 69% | 64% |
| **XGBoost** | 69% | 70% | 69% | 69% | 64% |
| **Catboost** | 69% | 70% | 69% | 69% | 64% |
| **Logistic Regression** | 66% | 68% | 66% | 67% | 63% |
| **KNN** | 64% | 68% | 64% | 65% | 63% |

*3.3. Individual Study Models*

Key results for the three Individual Study Models are given in Table 7.

For Study 1, the most accurate model used CatBoost, which was associated with a mean accuracy of 77%. This is a modest improvement on the chance level of accuracy which was 71%, based on the degree of class imbalance (i.e., 71% good responders). For Studies 2 and 3, the models did not provide an improvement on chance level.

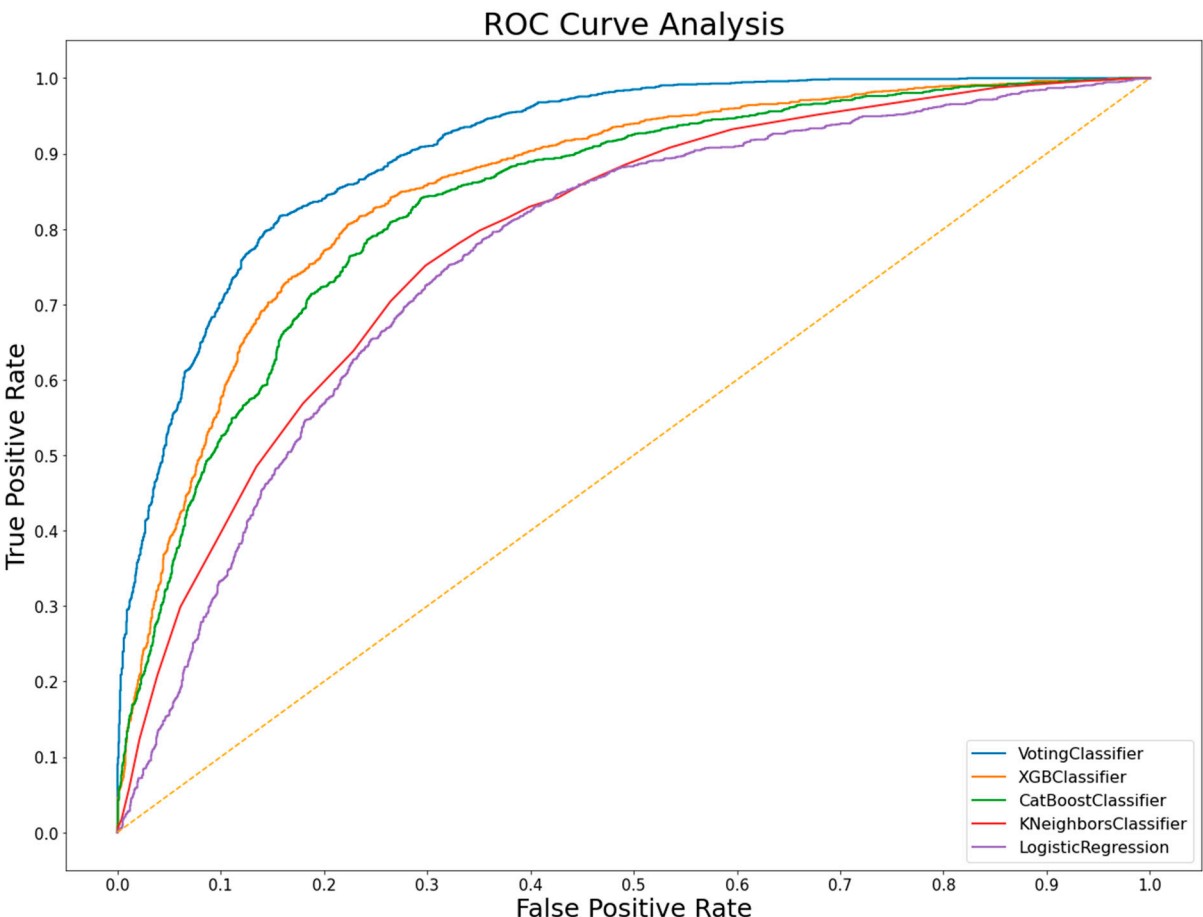

**Figure 2.** ROC curves for each algorithm, for the Three-Study Merged Dataset Models (using the full dataset for the fitted models).

**Table 7.** Performance Metrics (mean %s) for the Individual Study Models (averages are given for sensitivity, precision and F1 scores, weighted by the frequency of class labels).

|  |  | Accuracy | Precision | Sensitivity | F1 | AUC |
|---|---|---|---|---|---|---|
| **Study 1**<br>(chance level 71%) | **XGBoost** | 77% | 86% | 92% | 87% | 78% |
|  | **Voting classifier** | 75% | 84% | 97% | 95% | 77% |
|  | **Catboost** | 73% | 82% | 99% | 98% | 77% |
|  | **KNN** | 71% | 83% | 83% | 71% | 71% |
|  | **Logistic Regression** | 68% | 81% | 83% | 72% | 71% |
| **Study 2**<br>(chance level 70%) | **XGBoost** | 73% | 43% | 86% | 92% | 60% |
|  | **Voting classifier** | 71% | 42% | 99% | 100% | 52% |
|  | **Catboost** | 70% | 43% | 100% | 100% | 50% |
|  | **KNN** | 71% | 42% | 52% | 76% | 52% |
|  | **Logistic Regression** | 65% | 37% | 56% | 76% | 40% |
| **Study 3**<br>(chance level 84%) | **XGBoost** | 84% | 2% | 76% | 94% | 10% |
|  | **Voting classifier** | 83% | 7% | 100% | 100% | 18% |
|  | **Catboost** | 82% | 2% | 15% | 84% | 10% |
|  | **KNN** | 81% | 15% | 47% | 88% | 25% |
|  | **Logistic Regression** | 82% | 2% | 15% | 84% | 10% |

## 4. Discussion

We developed a series of ML models to predict two-year outcomes of a standard treatment for prostate cancer, based on baseline data from three clinical trials. This classification problem was challenging because it involved predicting responses a considerable

time in the future (two years). Despite this challenge, by creating a large dataset, merged from three clinical trials, we achieved an overall accuracy of 76% (the three-study merged dataset models). This level of accuracy represents a considerable improvement on the chance accuracy level of 54%. This level of accuracy also compares well with the only other study to use ML to predict clinical trial treatment outcomes (65% accuracy for responses to anti-depressant medication [15]). Most importantly, we suggest that this is the first time that accuracy levels have been achieved which are strong enough to form the basis of real-world applications. Our results are therefore a significant addition to the scientific literature and have important implications for both precision medicine and drug development strategy. Finally, this is the first study to use ML to predict individual treatment outcomes of clinical trials for cancer.

One crucial question is how far the predictive performance of such models can generalize to a real-world setting. To ensure the robustness of our results, we computed performance metrics by taking the mean of the results of a repeated (ten-fold) cross-validation procedure. More importantly, we assessed our models' generalizability by testing 'separate study validation models,' in which ML models were trained and tested on data from separate studies. This procedure provided a more stringent test of generalizability, for three reasons. First, differences between studies can compromise model accuracy. In our study there were small differences in entry criteria and there may have been unknown differences in the measurement of some variables. Second, such designs constrain the number of samples that can be used for model training and testing, which may introduce inefficiencies into the modeling. In our case, using data from Studies 1 and 3 for model training meant that this training set represented 65% of the total data. ML models are usually trained on higher proportions of the data than this, with 80% being typical. Therefore, the relatively low proportion of data available for model training likely compromised the accuracy of the separate study validation models. Third, the test set for the separate study validation models had a higher degree of class imbalance (30% good responders). Despite these challenges, the accuracy of the best model for the separate study validation models was 70%. This provides a strong confirmation of the robustness of our models and the feasibility of developing practical tools on the basis of such models, even where the quality of datasets is compromised by real-world limitations.

The real-world applicability of our predictive models is also underlined by that observation that the clinical trial data we used were fairly typical of Phase III clinical trials, in general, in terms of sample size and the kinds of assessments used (including data from questionnaires, biochemical assessments, clinical features, etc.). This points to the possibility that such methods could be applied not only to prostate cancer, or even cancer in general, but to many conditions for which Phase III clinical trial data is available.

For the three-study merged dataset models the most predictive feature was prior prostatectomy. In general, it can be difficult to infer the direction of the effects of features (e.g., whether prior prostatectomy increases or decreases the chance of being a 'good responder'), due to complex non-linearities in the relationships between predictors. However, in the case of prior prostatectomy, the difference in incidence between good and bad responders is so great (70% in good responders c.f., 37% in bad responders) that it can be confidently inferred that prior prostatectomy is associated with a *greater* chance that bicalutamide treatment will be successful at two years. However, our results do not necessarily indicate that prior prostatectomy 'causes' a better response to bicalutamide treatment, as this relationship may be indirect. For the other features, directional inferences are less safe because the differences between good and bad responders were much lower. Overall, the models can be considered to be essentially a 'black box,' i.e., it is predictive but may not provide clear information about how individual factors affect treatment success.

The individual study models were, in general, unsuccessful in predicting cases substantially above chance level. These results are unsurprising because, compared to the merged datasets, the single study datasets had smaller sample sizes and a substantial degree of class imbalance. Nonetheless, these results give an informal insight into the

relationships between modeling success, sample size, and class imbalance of the target variable. Such factors should be considered when planning the development of predictive tools using clinical trial data.

Our study aimed not to develop the most accurate model possible, which would involve the inclusion of more subjects and a more comprehensive range of data (genetics, proteomics, etc.). Instead, we aimed to develop ML-based tools for predicting cancer treatment outcomes based on data that would be practical and inexpensive to collect in a routine clinical setting. However, it would be interesting for further research to explore the effect of including other types of data that could improve predictive accuracy.

For ML-based systems to be effectively deployed in clinical practice, several predictive models would need to be developed so that a physician could compare treatment options. However, the quality and quantity of data available on the various treatment options would likely be variable, meaning that the accuracy of predictive systems would suffer from bias. Nonetheless, ML-based systems could still provide a major advance on the status quo, and such biases could be addressed by more systematic efforts to gather relevant data for modeling.

For both the three-study merged dataset models and the separate study validation models, the level of accuracy was similar for the three best performing models (for the three-study merged dataset models, these were 76% for XGB and voting classifier, and 75% with CatBoost). There were no statistically significant differences in accuracy between these models and thus their performance should be regarded as essentially equivalent.

The performance of the more sophisticated ML algorithms (CatBoost, XGBoost, and Voting Classification) was superior to that of the simpler algorithms (logistic regression and K-nearest neighbors). This likely reflects that the data included complex non-linearities, which tend to favor the more sophisticated ML algorithms [29].

There are several strengths of this study. First, we were able to construct models with a level of accuracy which could be feasible for real-world systems for precision medicine. Second, our models' robustness was demonstrated both by the use of internal cross-validation for testing and also by the use of separate studies for model training and testing. Third, we used a range of modeling approaches, both in terms of algorithms used and the procedures used to select training and testing data. These variations allow an examination of how such factors affect model performance. Fourth, the data we used were typical, in terms of assessments and sample sizes, of Phase III clinical trials. These considerations point to the feasibility of using Phase III clinical trial data to develop tools for real-world clinical practice.

Our study's most significant limitation is that it relied on a biomarker of disease progression (PSA levels) rather than direct measurements of drug efficacy, such as death or time to disease progression. A proxy for disease progression was necessary because there were insufficient data on death or disease progression for ML analysis. For this reason, the use of such biomarkers is common in clinical trials. A second limitation is that the measurement of some variables may have been suffered from biases between studies, for example, because the different studies were conducted in different countries. Such biases could have reduced the performance of the separate study validation models (where any such biases were not controlled) but do not pose a significant problem for the three-study merged dataset models (since the samples used for training and testing were shuffled) or the individual study models.

Despite its limitations, this study represents an important step in demonstrating the feasibility of ML-based tools for predicting cancer treatment outcomes, with implications for precision oncology and clinical trial enrichment.

**Author Contributions:** Conceptualization, F.D.B.; methodology, F.D.B., L.R.M.-P., S.G. and L.A.A.; formal analysis, F.D.B., S.G. and L.A.A.; data curation, F.D.B. and L.A.A.; writing—original draft preparation, F.D.B.; writing—review and editing, F.D.B. and L.R.M.-P.; visualization, F.D.B. and L.A.A. All authors have read and agreed to the published version of the manuscript.

**Funding:** This research received no external funding.

**Data Availability Statement:** Publicly available datasets were analyzed in this study. These datasets can be found in the following links: https://data.projectdatasphere.org/projectdatasphere/html/content/102 (accessed on 21 October 2020). https://data.projectdatasphere.org/projectdatasphere/html/content/105 (accessed on 21 October 2020); https://data.projectdatasphere.org/projectdatasphere/html/content/106 (accessed on 21 October 2020).

**Acknowledgments:** The authors thank D.R. Beacher and Giles Catcheside for their valuable comments on a draft of the manuscript.

**Conflicts of Interest:** The authors declare no conflict of interest.

## Appendix A. Outline of Study Design for the Three Studies Included in the Analyses Presented in This Paper

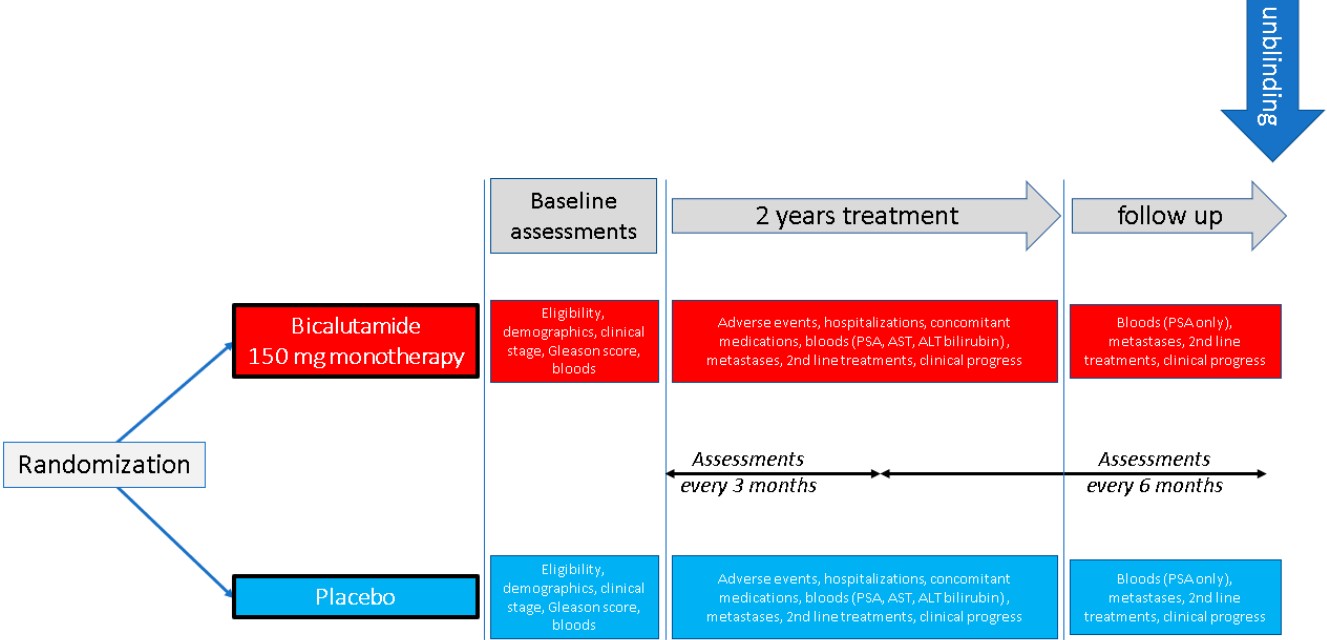

## Appendix B. Inclusion and Exclusion Criteria for the Three Astra-Zeneca EPC Studies

|  | Study 1 | Study 2 | Study 3 |
|---|---|---|---|
| **Inclusion criteria** | Adenocarcinoma of the prostate | | |
|  | Absence of metastatic disease | Non distant metastatic disease | |
|  | Radical prostatectomy OR prostate radiation treatment within 16 weeks before randomization. | 82% | 99% |
| **Exclusion criteria** | Any previous systemic therapy for prostate cancer except: 1. Neoadjuvant therapy prior to primary therapy 2. 5-alpha-reductase inhibitors | | |
|  | Other type of cancer (except treated skin carcinoma) within the last five years. | | Other type of cancer (except treated skin carcinoma) within the last ten years. |
|  | Serum bilirubin, AST or ALT >2.5x normal upper limit | | |
|  | Any severe medical condition that could jeopardize trial compliance | | |
|  | | Treatment with a new drug within previous 3 months | Patients for whom long term therapy is inappropriate due to low expected survival times |
|  | | | Patients at risk of transmitting infection through blood, including AIDS or other STDs or hepatitis |

## Appendix C. Target Variable and Frequencies/Means for the Features Included in the Three-Study Merged Dataset Models and Separate Study Validation Models

| Category | Feature | Study 1 | | Study 2 | | Study 3 | |
|---|---|---|---|---|---|---|---|
|  |  | Values | Missing | Values | Missing | Values | Missing |
| **Assessments used to compute 'good responders'** | Dropouts (<600 days of baseline) | 325 | 0 | 428 | 0 | 155 | 0 |
|  | PSA baseline (mean, ng/mL) | 0.65 | 62 | 11.5 | 24 | 24.8 | 0 |
|  | PSA 2 years (mean, ng/mL) | 0.6 | 139 | 22.5 | 305 | 32.6 | 112 |
|  | Prostate cancer death in 2y | 3 | 0 | 25 | 0 | 10 | 0 |
|  | Related A.E. led to withdrawal 2y | 73 | 0 | 76 | 0 | 17 | 0 |
|  | Related A.E. led to interventn 2y | 1 | 0 | 18 | 0 | 10 | 0 |
|  | Related A.E. led to hospitalizn 2y | 2 | 0 | 19 | 0 | 6 | 0 |
|  | Related A.E. led to disability 2y | 0 | 0 | 21 | 0 | 10 | 0 |
|  | Related A.E. life-threatening 2y | 0 | 0 | 12 | 0 | 3 | 0 |
|  | Any Serious Related A.E. in 2y | 74 | 0 | 98 | 0 | 21 | 0 |

| Category | Feature | Study 1 | | Study 2 | | Study 3 | |
|---|---|---|---|---|---|---|---|
| | | **Values** | **Missing** | **Values** | **Missing** | **Values** | **Missing** |
| **Demographics** | Age (mean years) | 64 | 0 | 69 | 0 | 68 | 0 |
| | Weight (mean kg) | 85 | 0 | 76 | 0 | 81 | 6 |
| | Clinical stage category | | | | | | |
| | Category 1 | 0 | 0 | 1 | 0 | 0 | 0 |
| | Category 2 | 23 | 0 | 153 | 0 | 44 | 0 |
| | Category 3 | 131 | 0 | 245 | 0 | 89 | 0 |
| | Category 4 | 1028 | 0 | 550 | 0 | 229 | 0 |
| | Category 5 | 440 | 0 | 446 | 0 | 213 | 0 |
| **Clinical features** | Category 6 | 3 | 0 | 45 | 0 | 13 | 0 |
| | Metastatic node disease | 462 | 0 | 41 | 0 | 24 | 0 |
| | Gleason score category | | | | | | |
| | Category 1 | 78 | 0 | 477 | 0 | 256 | 0 |
| | Category 2 | 788 | 0 | 574 | 0 | 266 | 0 |
| | Category 3 | 759 | 0 | 363 | 0 | 63 | 0 |
| | Category 4 | 0 | 0 | 26 | 0 | 3 | 0 |
| **Previous therapies for prostate cancer** | Prostatectomy | 1314 | 0 | 534 | 0 | 71 | 0 |
| | Radiotherapy | 319 | 0 | 295 | 0 | 30 | 0 |
| | PSA at baseline (mean, ng/mL) | 0.65 | 75 | 11 | 96 | 25 | 8 |
| | PSA within 1 year before baseline (mean, ng/mL) | 9.75 | 0 | 17.28 | 0 | 23.67 | 0 |
| **Biochemistry** | Baseline AST (mean, U/L) | 18 | 0 | 21 | 0 | 22.58 | 0 |
| | Baseline ALT (mean, U/L) | 20 | 0 | 22 | 0 | 21.72 | 0 |
| | Baseline Bilirubin (mean, UMOL/L) | 0.61 | 0 | 9.37 | 0 | 10.61 | 0 |

| Category | Feature | Study 1 | | Study 2 | | Study 3 | |
|---|---|---|---|---|---|---|---|
| | | Values | Missing | Values | Missing | Values | Missing |
| Generic names of concomitant medications | Amlodipine | 61 | 0 | 46 | 0 | 26 | 0 |
| | Aspirin | 385 | 0 | 247 | 0 | 76 | 0 |
| | Atenolol | 76 | 0 | 84 | 0 | 33 | 0 |
| | Diazepam | 18 | 0 | 10 | 0 | 11 | 0 |
| | Digoxin | 51 | 0 | 40 | 0 | 17 | 0 |
| | Doxazosin | 47 | 0 | 15 | 0 | 16 | 0 |
| | Enalapril | 64 | 0 | 82 | 0 | 28 | 0 |
| | Furosemide | 35 | 0 | 37 | 0 | 30 | 0 |
| | Glibenclamide | 55 | 0 | 39 | 0 | 16 | 0 |
| | Glyceryl trinitrate | 36 | 0 | 54 | 0 | 29 | 0 |
| | Lisinopril | 84 | 0 | 21 | 0 | 14 | 0 |
| | Metoprolol | 36 | 0 | 38 | 0 | 37 | 0 |
| | Nifedipine | 83 | 0 | 121 | 0 | 14 | 0 |
| | Omeprazole | 32 | 0 | 38 | 0 | 10 | 0 |
| | Paracetamol | 108 | 0 | 34 | 0 | 10 | 0 |
| | Salbutamol | 40 | 0 | 52 | 0 | 14 | 0 |
| | Simvastatin | 49 | 0 | 27 | 0 | 19 | 0 |
| | Vitamin, not otherwise specified | 255 | 0 | 14 | 0 | 0 | 0 |
| | Warfarin | 42 | 0 | 21 | 0 | 24 | 0 |

| Category | Feature | Study 1 | | Study 2 | | Study 3 | |
|---|---|---|---|---|---|---|---|
| | | **Values** | **Missing** | **Values** | **Missing** | **Values** | **Missing** |
| **PCLA drug class of concomitant medications** | Acetic acid derivs, rel substances | 47 | 0 | 41 | 0 | 12 | 0 |
| | Alpha-adrenoreceptor blockers | 114 | 0 | 55 | 0 | 18 | 0 |
| | Analgesics and antipyretics, anilides | 154 | 0 | 60 | 0 | 18 | 0 |
| | Analgesics and antipyretics, salicylic acid and derivs | 409 | 0 | 266 | 0 | 110 | 0 |
| | Angiotensin-converting enzyme inhibitors, plain | 230 | 0 | 192 | 0 | 59 | 0 |
| | Anti-anginal vasodilators, organic nitrates | 47 | 0 | 132 | 0 | 58 | 0 |
| | Antidiabetic sulfonamides urea derivs | 89 | 0 | 79 | 0 | 26 | 0 |
| | Anxiolytic benzodiazepine derivs | 65 | 0 | 74 | 0 | 19 | 0 |
| | Beta blocking agents, plain, non-Sel | 56 | 0 | 68 | 0 | 31 | 0 |
| | Beta blocking agents, plain, Sel | 114 | 0 | 153 | 0 | 78 | 0 |
| | Digitalis glycosides | 52 | 0 | 53 | 0 | 23 | 0 |
| | Glucocorticoids | 18 | 0 | 21 | 0 | 11 | 0 |
| | High-ceiling diuretics, sulfonamides, plain | 37 | 0 | 40 | 0 | 30 | 0 |
| | Hmg coa reductase inhibitors | 176 | 0 | 63 | 0 | 31 | 0 |
| | Inhal glucocorticoids | 39 | 0 | 74 | 0 | 21 | 0 |
| | Inhal Sel beta2-adrenoceptor agonists | 48 | 0 | 80 | 0 | 28 | 0 |
| | Low-ceiling diuretics and K-sparing agents | 32 | 0 | 44 | 0 | 14 | 0 |
| | Propionic acid derivs | 154 | 0 | 38 | 0 | 15 | 0 |
| | Proton pump inhibitors | 36 | 0 | 44 | 0 | 14 | 0 |
| | Sel Ca channel blockers, dihydropyridine derivs | 172 | 0 | 186 | 0 | 63 | 0 |
| | Vitamin k antagonists | 42 | 0 | 46 | 0 | 25 | 0 |

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
