# Peer review of "Machine Learning Predicts Outcomes of Phase III Clinical Trials for Prostate Cancer"

_algorithms, doi:10.3390/a14050147_

Round 1
Reviewer 1 Report
The paper is technically sound. The coverage of the topic is sufficiently comprehensive and balanced. The technical depth of the paper is appropriate for the generally knowledgeable individual Working in the Field.
Anyway, the results shown in the paper are OK.
A minor rewrite is required.
This is my advices to the authors:
In the abstract the authors show the goal of the paper, but what is their novelty with respect to literature?
I suggest writing it in the abstract.
I would propose a comparison of best performing models using machine learning (ML) to predict individual responses to a two-year course of bicalutamide from three different studies in time.
The database is well described.
The results are well commented and clear. The conclusions are clear and precise.
The References are significant and sufficient.
English language and style are clear.
Figures and tables are of sufficient resolution.
Author Response
"Please see the attachment."

Reviewer 2 Report
In this study, Beacher et al. proposed a machine learning approach to predict outcomes of phase III clinical trials for prostate cancer. Different conventional machine learning algorithms have been assessed on public datasets, thus the contribution of this study is limited. Some major issues are as follows:
1. Machine learning algorithms are well-known, datasets are public, so that what is the novelty of this study?
2. Overall, this manuscript looks like a project report, not scientific enough to be considered as an academic paper.
3. Performance results among different machine learning algorithms are similar in some algorithms. Therefore, how to select the optimal one is a question.
4. When comparing the predictive performance among different classifiers, the authors should perform some statistical tests to see the significant differences.
5. Did they tune all the optimal hyperparameters of all algorithms? It should be reported.
6. The study lack a comparison between their methods and previously published works on these datasets.
7. Source codes should be provided for replicating the methods.
8. The selected machine learning algorithms are well-known and have been used successfully in previous studies on biomedical data i.e., PMID: 32942564 and PMID: 33036150. Therefore, the authors are suggested to refer to more works in this description.
9. Quality of figures should be improved significantly.
10. Source codes should be provided for replicating the methods.
Reviewer 3 Report
The paper "Machine Learning Predicts Outcomes of Phase III Clinical Trials for Prostate Cancer" provides a first attempt in the prediction of outcomes of Phase III prostate cancer patients. The paper is overall well written and enjoyable. Too bad that the results are not very strong (i.e., above the chance level) but, again, for being a first attempt is fine.
The authors below can find my concerns.
Minor issues:
1. missing full stop after period on line 52-53
2. no closing parethesis on line 74-75, i.e. "(2013 U.S. Dollars [16]"
3. "case s" instead of "cases" on line 408
4. "false-positive" instead of "false positive" on line 407
Major issues:
1. mentioning "sklearn RobustScaler" on line 264 is inelegant. People that are not familiar with Python/Sklearn will not understand what this mean. The authors should explain in common terms how this scaler works so that people and understand and eventually replicate your experiments even in non-Python environments.
2. the "Derivation of the Target Variable" paragraph should be better explained. If I understood correctly, the classification problem deals with two classes. A patient can either be "good responder" or "bad responder", yet there is no mention of such information on such paragraph. The authors should stress that if a subject does not met the 4 conditions it is marked as "bad responder".
3. the "Final Datasets" paragraph also needs some intervention. The authors say that "The individual study datasets were highly imbalanced in terms of the classes for the target variable ('good responders')". Although this is true only for Study 1. Studies 2 and 3 are imbalanced in terms of 'bad responders' since there are only 30% and 16% of 'good responders'.
4. in "Separate Study Validation Models" (line 301), the authors should better motivate the choice behind Study 2 as separate test set. In fact, the authors say that Study 2 has a good balance between 'good responders' (30%, from Table 3) and 'bad responders' (100%-30%=70%). However, the same balance ratio holds for Study 1, which features 71% of 'good responders' and 29% of 'bad responders'. The positive-vs-negatives ratio is the same, although I understand that in Study 1 'good responders' are the majority and in Study 2 'bad responders' are the majority.
5. in Section 3.1, it is again inelegant to use Python nomenclature, that is "n_estimators=300, max_depth=5, learning_rate=0.01, base_score=0.5, 384 booster='gbtree', min_child_weight=5, reg_lambda=0.8, reg_alpha=0.3, colsam- 385 ple_bytree=0.5, subsample=0.8." People that are not familiar with Python/Sklearn will not understand what these items mean. This point somehow links to my point #1
6. Caption of Table 6 (and other tables). The authors say that the sensitivity, precision and F1 are shown as weighted averages. "Weighted" against what?
7. If I understood correctly, the "chance level" is evaluated by accounting the percentage of positive instances in the dataset (e.g., if I have 10 positive instances across 100 samples, the chance level is 10%). If this is true, why in Table 7, for Study 2, the chance level is 70% whereas the number of 'good responders' in Table 3 for Study 2 is 30%?
8. In order to tune the hyperparameters (grid/random search, 10-fold cross-validation) which performance metric has been optimised? The classification accuracy? It is worth specifying.
Round 2
Reviewer 2 Report
My previous comments have been addressed well.
Reviewer 3 Report
The authors did a very good job in addressing my concerns and revising the manuscript, which I think is now ready for publication.